# Reproductive number of coronavirus: A systematic review and meta-analysis based on global level evidence

**Md. Arif Billah**[1], **Md. Mamun Miah**[2], **Md. Nuruzzaman Khan**[3]*

1 Faculty of Business, Economic and Social Development, University Malaysia Terengganu, Terengganu, Malaysia, 2 Department of Mathematics, Khulna University of Engineering and Technology, Khulna, Bangladesh, 3 Department of Population Science, Jatiya Kabi Kazi Nazrul Islam University, Trishal, Mymensingh, Bangladesh

* sumonrupop@gmail.com

**Data Availability Statement:** This is systematic review of the relevant published papers. All included papers results are reported in the manuscript and its supplementary file.

## Abstract

### Background

The coronavirus (SARS-COV-2) is now a global concern because of its higher transmission capacity and associated adverse consequences including death. The reproductive number of coronavirus provides an estimate of the possible extent of the transmission. This study aims to provide a summary reproductive number of coronavirus based on available global level evidence.

### Methods

A total of three databases were searched on September 15, 2020: PubMed, Web of Science, and Science Direct. The searches were conducted using a pre-specified search strategy to record studies reported the reproductive number of coronavirus from its inception in December 2019. It includes keywords of coronavirus and its reproductive number, which were combined using the Boolean operators (AND, OR). Based on the included studies, we estimated a summary reproductive number by using the meta-analysis. We used narrative synthesis to explain the results of the studies where the reproductive number was reported, however, were not possible to include in the meta-analysis because of the lack of data (mostly due to confidence interval was not reported).

### Results

Total of 42 studies included in this review whereas 29 of them were included in the meta-analysis. The estimated summary reproductive number was 2.87 (95% CI, 2.39–3.44). We found evidence of very high heterogeneity (99.5%) of the reproductive number reported in the included studies. Our sub-group analysis was found the significant variations of reproductive number across the country for which it was estimated, method and model that were used to estimate the reproductive number, number of case that was considered to estimate the reproductive number, and the type of reproductive number that was estimated. The highest reproductive number was reported for the Diamond Princess Cruise Ship in Japan

**Funding:** The author(s) received no specific funding for this work.

**Competing interests:** The authors have declared that no competing interests exist.

**Abbreviations:** CCDC, Chinese Center for Disease Control and Prevention; COVID-19, Novel Coronavirus Disease 2019; CI, Confidence Interval; EGR, Exponential Growth Rate; GT, Generation Time; MCO, Movement Control Order; MCMC, Markov Chain Monte Carlo; MERS, Middle East Respiratory Syndrome; MLE, Maximum Likelihood Estimation; nCoV-19, Novel Coronavirus 2019; NGMA, Next Generation Matrix Approach; NIH, National Institute of Health; NP, Non-Pharmaceutical; PRISMA, Preferred Reporting Items for Systematic Reviews and Meta-Analyses; SARS, Severe Acute Respiratory Syndrome; SARS-COV-2, Severe Acute Respiratory Syndrome due to Coronavirus-2; SEIQ, Susceptible, Exposed, Infected, Quarantined; SEIHR, Susceptible, Exposed, Infected, Hospitalized, Removed/Recovered; SEIR, Susceptible, Exposed, Infected and Removed/Recovered; SI, Serial Interval; SIR, Susceptible, Infected and Removed/Recovered; USA, United States of America; WHO, World Health Organization.

(14.8). In the country-level, the higher reproductive number was reported for France (R, 6.32, 95% CI, 5.72–6.99) following Germany (R, 6.07, 95% CI, 5.51–6.69) and Spain (R, 3.56, 95% CI, 1.62–7.82). The higher reproductive number was reported if it was estimated by using the Markov Chain Monte Carlo method (MCMC) method and the Epidemic curve model. We also reported significant heterogeneity of the type of reproductive number- a high-value reported if it was the time-dependent reproductive number.

## Conclusion

The estimated summary reproductive number indicates an exponential increase of coronavirus infection in the coming days. Comprehensive policies and programs are important to reduce new infections as well as the associated adverse consequences including death.

## Background

Coronavirus (SARS-COV-2) is now a global concern that speared out to 213 countries or territories as of September 15, 2020. More than 29.5 million population have been infected so far worldwide, of which more than 933,720 are died [1]. Consequently, the World Health Organization (WHO) has declared it as pandemic and suggested countries to take aggressive measures to reduce new infections [2]. Given no treatments or vaccines available for this virus, countries are now imposing numerous non-medical measures to reduce further infections, which include restricting people's movements, banned international and local travels, quarantine, and isolation [3]. However, the new infections are rising exponentially, in all ages and sexes, irrespective of the countries [4, 5]. Reducing new infections, therefore, needs further comprehensive preventive measures.

Knowing the accurate reproductive number of coronavirus, defined as the capability of transmission per primary infected person to the secondarily infected persons, is significant for various reasons, including to assess epidemic transmissibility and to predict the future trend of spreading [6]. These are important to reduce new infections through designing effective control measures such as social distancing [7] and to know the expected duration of keeping control measures [5]. Moreover, it also helps to develop an effective epidemiological mathematical model considering possible transmission ways, such as, droplets and direct contacts with coronavirus infected patients (COVID-19), which are important to know the risk population and the appropriate epidemiologic parameters [8, 9].

There are various researches in the country level that have been reported the reproductive number of coronavirus. However, they were not consistent in terms of their measurement procedures and methods used, therefore, the estimated reproductive number was quite different [8, 10]. Other reported sources of variations of the reported reproductive number were the country for which the reproductive number was estimated and its stages of infection and preventive measures applied [11]. Another important source of variation of the estimated reproductive number was the type of reproductive numbers considered [8]. Of the three reproductive numbers estimated, namely the basic reproductive number ($R_0$), net reproductive number ($R_e$), and time dependent reproductive number ($R_t$), are applicable for different purposes. For instance, the basic reproductive number is used when an infected person can mix randomly to non-infected persons (i.e., no control intervention was applied), whereas, the net and time-dependent reproductive number are used when control interventions were applied.

To settle these disagreements on the reported reproductive number and know the current situation of infection, a summary estimate of the reproductive number is important. However, of the three studies that have been provided summary reproductive number so far were limited in several areas and did little to settle these disagreements [12–14]. For instance, they reported a summary estimate of the basic reproductive number without considering the net reproductive number and the time-dependent reproductive number. However, it is around 10 months that have already been gone since the first infection of the coronavirus in December 2019 and all countries have been imposed several prevention measures. Therefore, the estimation of the basic reproductive number was available only in a few studies of which these summary estimates were based. Moreover, these studies were also failed to address the heterogeneity of their estimated reproductive number though it was found higher [12–14].

Considering the higher variability of the reported reproductive number and lack of relevant research, in this study, an attempt has been made to provide a summary reproductive number of coronavirus. The sources of variation of the reported reproductive number were also addressed. Findings will help policymakers to know about the possible increase of coronavirus infected patients and take policies and programs accordingly.

## Methods

Literature searches were conducted in three databases on September 15, 2020: PubMed, Web of Science, and Science Direct. The pre-specified search strategies were used to search databases (S1-S3 Tables in S1 File). We developed search strategies consisting of virus-specific (corona virus, coronavirus, SARS-CoV-2, COVID-19, nCoV-2019) and reproductive number related (reproduction number, transmissibility) keywords that were combined using the Boolean operators (AND, OR). Additional searches were conducted in the reference list of the selected articles, and the relevant journal's websites.

### Inclusion and exclusion criteria

Studies meet the following inclusion criteria were included: wrote in the English language, presented a reproductive number of the coronavirus instead of considering its type (basic reproductive number, net reproductive number, and time-dependent reproductive number. We did not apply any time restriction, i.e. all studies from the onset of coronavirus to the date of conducting formal search were included. Studies that did not meet these criteria were excluded.

### Data extraction

Two authors (MAB, MMM) extracted information by using a pre-designed, trailed, and modified data extraction sheet. The extracted information includes: year of publication, study's location, model used to estimate the reproductive number, time for when the reproductive number was estimated, number of cases considered to estimate the reproductive number, assumption(s) that was/were set to a calculate the reproductive number, intervention strategy, and the estimated reproductive number with its 95% confidence interval (CI). The corresponding author (MNK) solved any disagreement on information extraction.

### Statistical analysis

The information recorded were mostly dichotomous where the numerical reproductive number was reported in all selected studies. We, therefore, used both narrative synthesis and meta-analysis to summaries findings from retrieved studies. Narrative synthesis was used to explain the findings of the studies where the reproductive number was reported, however, its 95%

confidence interval was missing that did not enable them to be included in the meta-analysis. Meta-analysis was used for the studies that consistently reported the reproductive number and its 95% confidence interval. We first use the fixed-effect meta-analysis to get a pool reproductive number for the studies which reported more than one reproductive number for a country calculated based on different assumptions. Later this pooled estimate was used to give a summary estimate of the reproductive number. We used the random-effect meta-analysis to estimate the summary reproductive number. The model was chosen based on the heterogeneity assessment ($I^2$) which reported a very high heterogeneity of the reported reproductive number across different included studies. Later we explored the sources of heterogeneity through subgroups analysis across the selected studies' characteristics. These include the country for which the reported reproductive number was estimated, the method and model that were used to estimate the reproductive number, total number of case that was considered to estimate the reproductive number and type of reproductive number that was reported. We also assessed the publication bias through visual inspection of the funnel plot and Egger's regression asymmetry test. The trim-and-fill procedure was used when evidence of publication bias was found. The National Institutes of Health (NIH) study quality assessment tool was used to assess study quality. The Stata software version 15.1 (Stata Corp, College Station, Texas, USA) was used to perform all analyses.

## Results

### Literature search results

Total of 541 studies included, 528 of them were extracted from three databases searched (Fig 1 and S1-S3 Tables in S1 File). Of these, 494 studies were excluded through title and abstract screening leaving 47 studies for full-text review. A total of 42 of them were finally included in this study and 29 of them were included in the meta-analysis. All included studies were moderate to high in quality (Table 1 and S4 Table in S1 File).

Majority of the studies selected were conducted in China (8) [6, 15–21] and its province (6) [22–27]. The remaining studies were conducted in Japan (3) [28–31] followed by South Korea (3) [32–34], Italy (2) [35, 36], Spain (2) [36, 37], and France and Germany (1) [36]. Four studies included were conducted based on multiple countries' data [7, 38–40].

### Estimated reproduction number

The estimated summary reproductive number based on the 29 studies included in the meta-analysis was 2.87 (95% CI, 2.39–3.34) (Fig 2). We found a very high heterogeneity (99.5%) of the reported reproductive number of these included studies. However, we did not find any evidence of publication bias (Fig 3). We used the subgroup analysis to address the heterogeneity of the reported reproductive number across selected studies characteristics. Their results are reported in Table 2 and the details results are presented in the S1-S5 Figs in S1 File. We found heterogeneity of the reported reproductive number across the countries for which the reproductive number were estimated, models and methods that were used to estimate the reproductive number, and the total number of cases that was used to estimate the reproductive number, and the type of the reproductive numbers that were estimated. For instance, the estimated reproductive number was higher in outside of China (R, 4.56, 95% CI, 2.28–9.12) than the mainland of China (R, 3.14, 95% CI, 2.40–4.09). However, in the country level, the highest reproductive number was reported for France (R, 6.32, 95% CI, 5.72–6.98) following Germany (R, 6.07, 95% CI, 5.51–6.69) and Spain (R, 5.08, 95% CI, 4.50–5.73). South Korea was the only country reported <1 reproductive number (R, 0.76, 95% CI, 0.34–1.70). The higher reproductive number reported if it was estimated by the Markov Chain Monte Carlo method (MCMC)

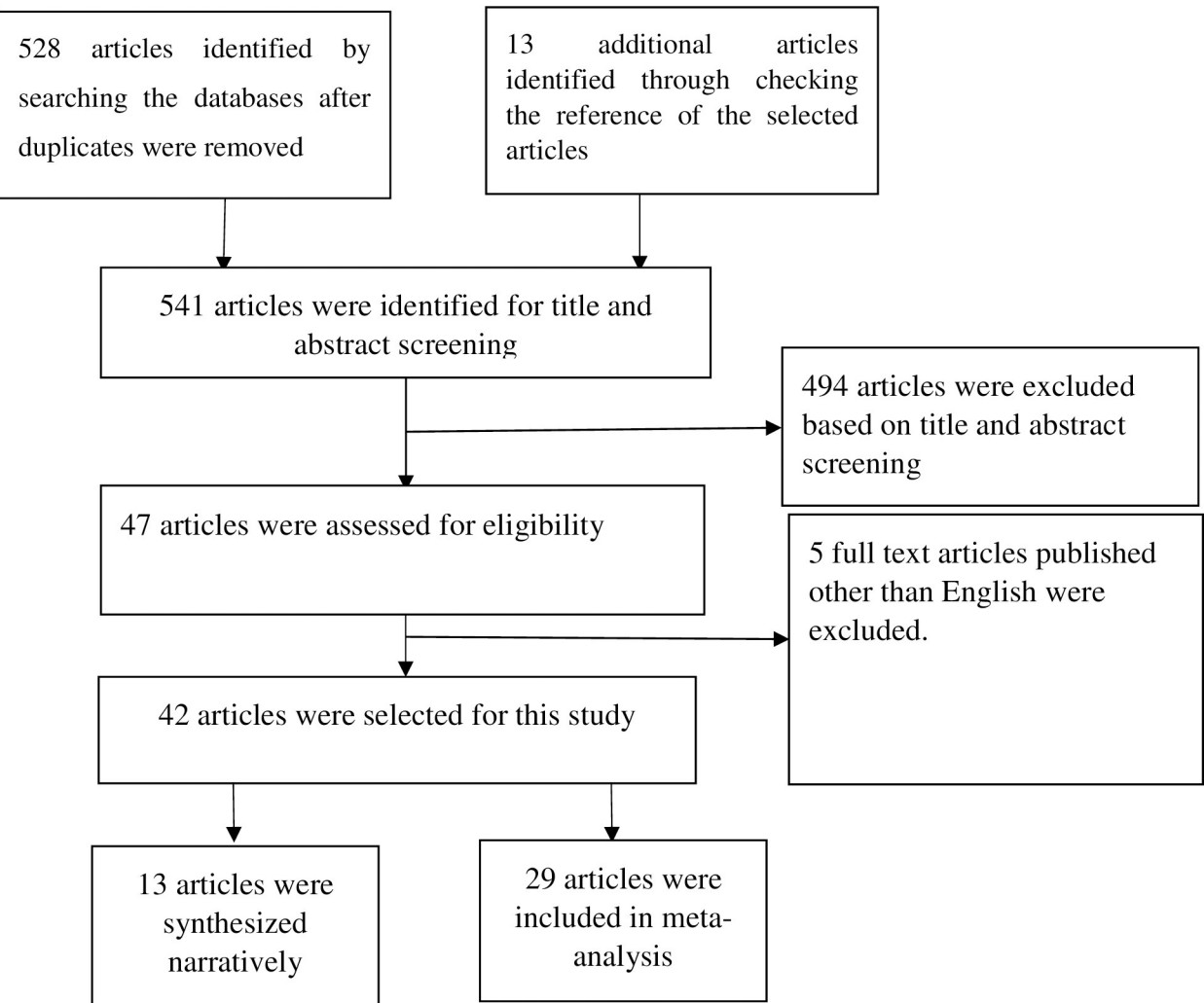

**Fig 1. Schematic representation of the included studies reporting the reproductive number of coronavirus published between December 2019 and September 2020.**

method (R, 4.57, 95% CI, 2.68–7.78) and by the Epidemic curve model (R, 3.04, 95% CI, 2.60–3.55). The summary reproductive number was found higher if it was estimated for >3162 cases (R, 3.27, 95% CI, 2.47–4.31) than ≤3162 cases (R, 2.51, 95% CI, 1.91–3.28). Variations were also found across the type of reported reproductive numbers- the time-dependent reproductive number was found around double (R,4.42; 95% CI, 3.05–6.40) than the net reproductive number (R,1.95; 95% CI, 1.63–2.34). However, we found, through using the meta-regression, these differences were only significant across the countries of the reported reproductive number and the methods used to estimate the reproductive number.

The results of the 13 studies that are narrative synthesized are presented in Table 3. Their findings were in line with our estimated summary reproductive number. Only a study conducted for Diamond Princes Cruise Ship, Japan reported a very high reproductive number, 14.8 for the period of 21 January to 19 February 2020 [30]. However, this estimated reproductive number was conditioned for not to be applied any preventive intervention and the infected person can mix randomly to the non-infected persons. When preventive interventions applied this number was reduced to 1.78.

**Table 1. Background characteristics of the 29 studies included in the meta-analysis.**

| Serial number | Author, Study's location | Model used to estimate the reproductive number | Time/period for which the reproductive number was estimated | Assumption(s) that was/ were considered to estimate the reproductive number | Method used to estimate the reproductive number | Reproductive number (95% CI) | Study-quality assessment (earned score in the scale of 9)[++] |
|---|---|---|---|---|---|---|---|
| 1 | Read et al, 2020 [40], China and overseas | Susceptible-Exposed-Infected-Removed (SEIR) model | 1st Jan 2020 to 22nd Jan 2020 | Cases daily time increase follows a Poisson distribution | MLE[1] | 3.11 (2.39–4.13) | 7 |
| 2 | Zhang et al., 2020 [31], Diamond Princess Cruise ship, Japan | Epidemic model incorporated by the data | 16th Feb 2020 | The mean serial interval (SI)[2] 7.5 days, standard deviation (SD) 3.4 days | MLE | 2.28 (2.06–2.52) | 7 |
| 3 | Liu et al., 2020 [39], China and overseas | No model mentioned | before 23rd Jan 2020 | With generation time (GT)[3] of 8.4 days | EGR | 2.90 (2.32–2.52) | 8 |
| | | | | | MLE | 2.92 (2.28–3.67) | |
| 4 | Majumder & Mandl, 2020 [24], Wuhan, China | Susceptible-Infected-Recovered/ Removed (SIR) model | Dec 8, 2019, to Jan 26, 2020 | Mean SI 8 (range 6–10) days | SEIR method | 2.55 (2.00–3.10) | 6 |
| 5 | Riou & Althaus, 2020 [7], China and overseas | No model mentioned | before 18th Jan 2020 | The mean GT varied 7–14 days | Stochastic simulation | 2.2 (1.4–3.8) | 8 |
| 6 | Tang et al., 2020 [18], China | SEIR model (with isolation, quarantined) | 31 Dec 2019 to 15th Jan 2020 | The incubation period is 7 days | NGMA[1] | 6.47 (5.71–7.23) | 9 |
| 7 | Zhao, Lin et al., 2020 [19], China | Epidemic curve by time-series data | 10th Jan to 24th Jan 2020 | 8-fold reporting rate | EGR | 2.24 (1.96–2.55) | 7 |
| | | | | 2-fold reporting rate | | 3.58 (2.89–4.39) | |
| | | | | 0-fold reporting rate | | 5.71 (4.24–7.54) | |
| 8 | Zhao, Musa, et al., 2020 [20], China | Epidemic curve using time series information | 1st Jan to 15th Jan 2020 | Constant screening effort applied in the Wuhan at the same point in time. | EGR | 2.56 (2.49–2.63) | 8 |
| 9 | Shen et al., 2020 [25], Hubei province, China | SEIR model | 12th Dec 2019 to 22nd Jan 2020 | 5–6 days of incubation | SEIR method[1] | 4.71 (4.50–4.92) | 8 |
| | | | | With intervention and 5–6 days of the incubation period | SEIR method | 2.08 (1.99–2.18) | |
| 10 | Q. Li et al., 2020 [23], Wuhan, China | Epidemiologic time delay distribution | Before 22nd Jan 2020 | Mean SI 8.4 days and SD 3.8 days | Fitted transmission model with zoonotic infection | 2.20 (1.40–3.90) | 8 |
| 11 | J. T. Wu et al., 2020 [27], Wuhan, China | SEIR model | 31 Dec 2019 to 28th Jan 2020 | Mean SI of 8.4 days | MCMC[1] | 2.68 (2.47–2.86) | 9 |
| 12 | Imai et al., 2020 [15], China | No model mentioned | before 18th Jan 2020 | High level of variability & generation time is 8.4 days | Computational modelling epidemiologic trajectories | 2.60 (1.50–3.50) | 7 |
| 13 | Kucharski et al., 2020 [38], Wuhan and international travellers | SEIR model | 29th Dec 2019 to 23rd Feb 2020 | Mean incubation period is assumed to be 5.2 days & SD 3.7 days | MLE | 2.35 (1.15–4.77) | 9 |
| | | | | Intervention with mean incubation period 5.2 days & SD 3.7 days | MLE | 1.05 (0.41–2.39) | |
| 14 | Ki, 2020 [33], South Korea | Epidemic curve fitting | 20 Jan to 10 Feb 2020 | Not Available (NA) | EGR | 0.48 (0.25–0.84) | 9 |
| 15 | Choi & Ki, 2020 [32], South Korea | SEIR model | 20 Jan to 17 Feb, 2020 | Overseas infections are separated | SEIR method | 0.56 (0.51–0.60) | 9 |

(*Continued*)

**Table 1.** (Continued)

| Serial number | Author, Study's location | Model used to estimate the reproductive number | Time/period for which the reproductive number was estimated | Assumption(s) that was/were considered to estimate the reproductive number | Method used to estimate the reproductive number | Reproductive number (95% CI) | Study-quality assessment (earned score in the scale of 9)[++] |
|---|---|---|---|---|---|---|---|
| 16 | Shim et al., 2020 [34], South Korea | Epidemic curve fitting with the growth model | 20th Jan to 26th Feb 2020 | With mean GT 4.41 days and SD 3.17 days | Simulation | 1.50 (1.40–1.60) | 8 |
| 17 | Lai et al., 2020 [41], Genetic data from GISAID | Phylogenetic estimation | 4th Feb 2020 | Based on the exponential growth rate of 0.218 per days | EGR | 2.60 (2.10–5.10) | 9 |
| | | | | The evolutionary rate set to the value of $8.0 \times 10{-4}$ subs/site/year | Birth-death skyline estimate | 1.85 (1.37–2.40) | |
| 18 | Jung et al., 2020 [42], Outside of China | No model mentioned | before 24 Jan 2020 | Mean SI 7.5 days and SD 3.4 days | EGR | 3.19 (2.66–3.69) | 8 |
| 19 | Song et al., 2020 [17], China | SEIR model | 15 to 31 Jan 2020 | Using generation intervals | EGR | 3.74 (3.63–3.87) | 6 |
| | | | | Using generation intervals | MLE | 3.16 (2.90–3.43) | |
| | | | | The model fitted best 27th Jan | SEIR method | 3.91 (3.71–4.11) | |
| 20 | Sanche et al., 2020 [16], China | SEIR model | 15 to 30 Jan 2020 | with 7–8 days of the SI | EGR | 5.80 (4.40–7.70) | 7 |
| | | | | with 6–9 days of the SI | | 5.7 (3.80–8.90) | |
| 21 | Mizumoto & Chowell, 2020 [29], Diamond Princes Cruise ship, Japan | No model mentioned | 20 Jan to 18 Feb, 2020 | Mean SI 7.5 days and SD 3.4 | NGMA | 5.8 (0.6–11.0) | 9 |
| 22 | Kuniya, 2020 [28], Japan | SEIR model | 15 Jan to 29 Feb 2020 | Infected increases at a rate of daily time increment | NGMA | 2.60 (2.40–2.80) | 6 |
| 23 | Iwata & Miyakoshi, 2020 [43], Outside of China | SEIR model | Not Available (NA) | One infected entered a community of 1000 population. | MCMC | 6.5 (5.6–7.2) | 7 |
| 24 | Wan et al., 2020 [26], Wuhan, China | SEIR model | 22 Jan to 07 Feb 2020 | 7 days incubation period and 14 days of the infectious period | SEIR method | 1.44 (1.40–1.47) | 8 |
| 25 | Yuan et al., 2020 [36], Italy | No model mentioned | 23 Feb to 9 Mar 2020 | Mean GT 5.6 days and SD 2.6 days | EGR | 3.27 (3.17–3.38) | 9 |
| | Yuan et al.,2020 [36], France | | | | | 6.32 (5.72–6.99) | |
| | Yuan et al.,2020 [36], Germany | | | | | 6.07 (5.51–6.69) | |
| | Yuan et al.,2020 [36], Spain | | | | | 5.08 (4.51–5.74) | |
| 26 | Chintalapudi et al., 2020 [44], Italy | No model | 26 Feb to 20 Apr 2020 | Using estimated SI with non-pharmaceutical (NP) interventions | MLE | 1.85 (0.60–2.30) | 8 |
| 27 | Hyafil and Morina, 2020 [37], Spain | SIR model | Upto 13 Mar 2020 | Based on the hospitalized data with 7.65 days incubation period | SEIR method | 5.89 (5.86–7.09) | 8 |
| | Hyafil and Morina, 2020 [37], Spain | SIR model | Upto 13 Mar 2020 | Based on the detected cases with 10.2 days incubation period | SEIR method | 6.91 (6.95–7.39) | |
| | Hyafil and Morina, 2020 [37], Spain | SIR model | 16 Mar to 15 Apr 2020 | Based on the hospitalized data with 7.65 days incubation period with initial interventions | SEIR method | 1.86 (1.10–2.63) | |

(*Continued*)

**Table 1.** (Continued)

| Serial number | Author, Study's location | Model used to estimate the reproductive number | Time/period for which the reproductive number was estimated | Assumption(s) that was/ were considered to estimate the reproductive number | Method used to estimate the reproductive number | Reproductive number (95% CI) | Study-quality assessment (earned score in the scale of 9)[++] |
|---|---|---|---|---|---|---|---|
| | Hyafil and Morina, 2020 [37], Spain | SIR model | 16 Mar to 15 Apr 2020 | Based on the detected cases with 10.2 days incubation period with initial interventions | SEIR method | 2.22 (1.92–2.74) | |
| | Hyafil and Morina, 2020 [37], Spain | SIR model | 31 Mar to 12 Apr 2020 | Based on the hospitalized data with 7.65 days incubation period with interventions for full restrictions | SEIR method | 0.48 (0.15–1.17) | |
| | Hyafil and Morina, 2020 [37], Spain | SIR model | 31 Mar to 12 Apr 2020 | Based on the detected cases with 10.2 days incubation period with interventions for full restrictions | SEIR method | 0.85 (0.50–1.05) | |
| 28 | Zhang et al., 2020 [45], Wuhan, China | SEIQ model | 21 Jan to 20 Feb 2020 | Mean SI 5.2 days and hospital quarantine 12.5 days | MCMC | 5.50 (5.20–5.80) | 7 |
| 29 | Shao et al., 2020 [46], China | Fiduan-CCDC model | Not specified | Mean SI 7.5 days with SD 3.4 days | SEIR method | 3.32 (3.25–3.40) | 7 |

Note: All studies included in the meta-analysis were summarized in this table. Studies included in the narrative synthesis were summarized in Table 3. [1]EGR: Exponential growth rate method; MLE: Maximum Likelihood Method; MCMC: Markov Chain Monte Carlo Method; NGMA: Next-Generation Matrix Approach and SEIR method = $\beta/\gamma$ method. R: Reproductive number, 95% CI, 95% Confidence Interval.

[2]Serial interval refers to the duration of time between the onset of symptoms in an index case and a secondary case.

[3]Generation time refers to the time interval between successive infections in the chain of transmission.

[++]Study quality was assessed through the National Institutes of Health (NIH) study quality assessment. Details results are presented in S4 Table in S1 File.

## Discussion

This review aimed to provide the summary reproductive number of the coronavirus based on the global level evidence. A total of 42 studies selected for this study of which 29 studies were included in the meta-analysis. Majority of the included studies were conducted in China. The estimated summary reproductive number was 2.87. We found evidence of higher heterogeneity of the reported reproductive number across different studies. The sources of heterogeneity were the country for which the reproductive number was estimated, models and methods that were used to estimate the reproductive number, and the total number of case that was used to estimate the reproductive number.

The average estimated reproductive number was 2.87; which is higher than the WHO's estimate of 1.4 to 2.5. However, this estimate is lower than the previous summarized reproductive number of coronavirus, 3.38 estimated by Alimohamadi and Colleagues based on the 23 studies [12], 3.15 reported estimated by He and colleagues [14] based on the 7 studies, and 3.28 estimated by Liu and colleagues based on the included 12 studies [13]. Numerous measures to reduce new infections of coronavirus such as social distancing, and controlling international travels are associated with such reduction [54, 55]. However, our estimated reproductive number is still very high that could have the potential to an exponential increase in new infections. Moreover, the estimated number is still very higher than previous rounds of coronavirus like infectious diseases, such as severe acute respiratory syndrome (SARS) and the Middle East respiratory syndrome (MERS) if we considered the period between the when was estimation

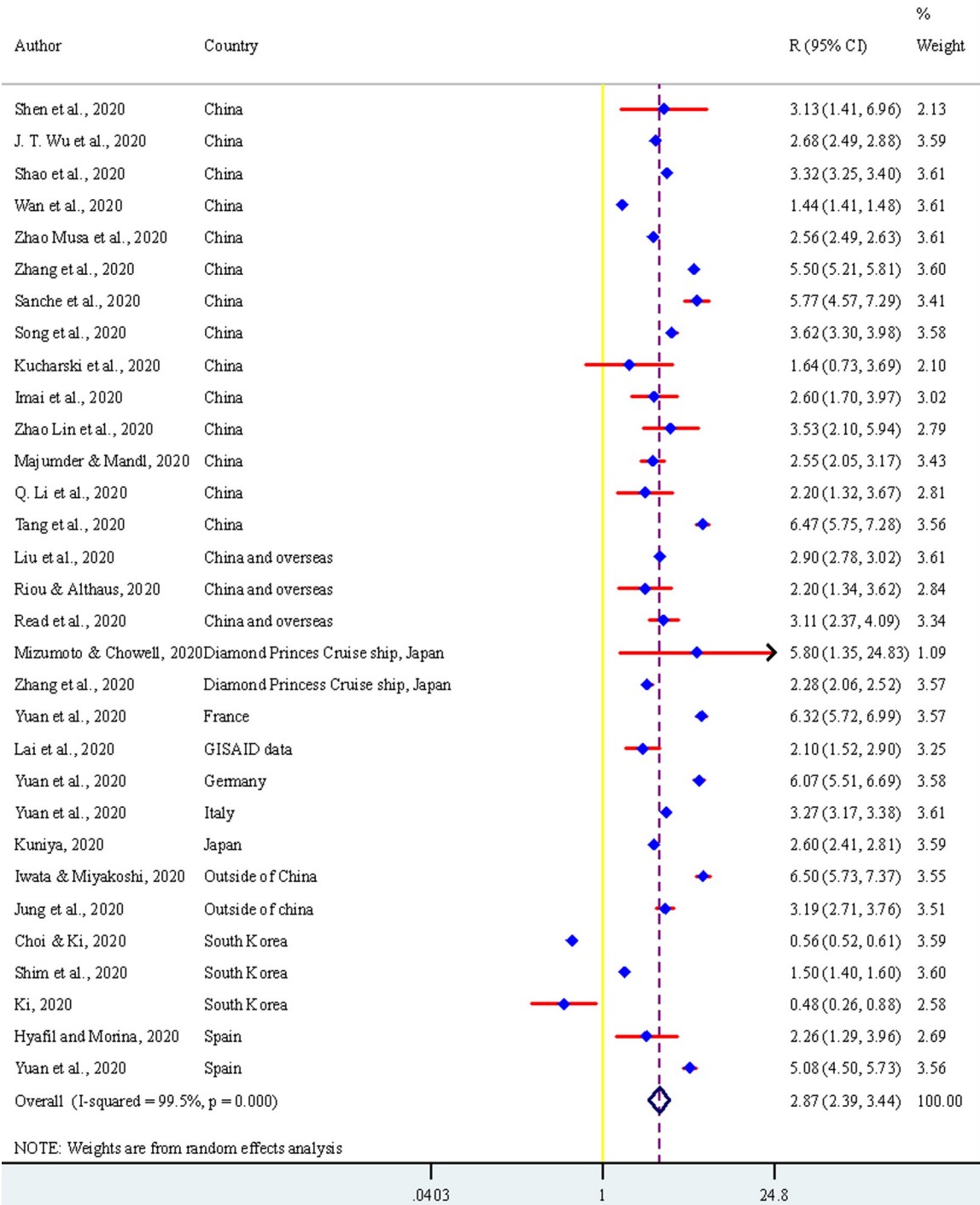

**Fig 2. Estimated summary reproductive number of coronavirus based on 29 studies with 32 times report.** Note: One study [36] reported estimates for four different countries: France, Germany, Italy, and Spain.

done and infections were initially detected. For instance, the reproductive numbers of SARS and MERS were reduced to 0.95 (95% CI, 0.61–1,23) and 0.91 (95% CI, 0.36–1.44), respectively, after 3rd generation of the infection [56]. There are numerous reasons for such a higher reproductive number. First, biological facts of the infection rate and duration of contagion are

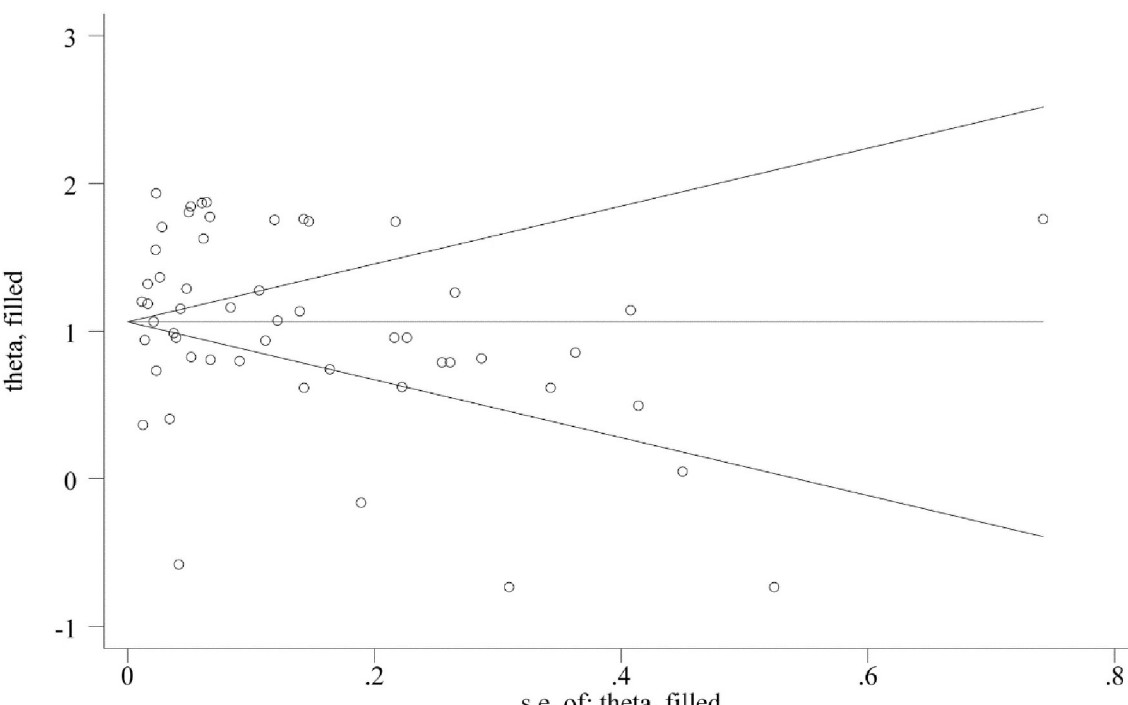

**Fig 3. Funnel plot to identify publication bias including all studies used to estimate the summary reproductive number of coronavirus (Egger test p-value, 0.556).**

important to explain such higher reproductive number instead of strict control measures that placed to reduce new infections [57]. For instance, a person could be infected in numerous ways, such as gets physically contacted with the infected person or through environmental transmission by respiratory droplets [58]. Moreover, coronavirus infected patients may not show symptomatic characteristics upto two weeks of infection. This pre-symptomatic stage is another vital source to increase new infections exponentially as in this period an infected person is usually confounded in the community with other people. This risk is further increased significantly for the country where population density is high [59].

This study also found evidence of the very high (99.5%) heterogeneity of the estimated reproductive number. Along with the factors described above, the study's characteristics were found as the important sources for such higher heterogeneity. For instance, the reproductive number found higher for the countries where no restriction was applied, or restriction was applied in delayed. The forms of restrictions were to control people's movement, to monitor personal hygiene, and to impose to wearing a mask [60, 61]. These implications act to control virus transmission from an infected to the susceptible and reduce the new infections. These also affect the average transmissibility of coronavirus within the specific population and settings [62, 63].

Estimation models, assumptions applied, and estimation processes were empirical sources of variability of the estimated reproductive number of coronavirus [64]. For instance, studies included in this analysis were followed assumption of generation time (which is followed by the gamma distribution) or serial interval (which is followed by the poison distribution) which is an important source of heterogeneity [65–67]. The reason of such difference is the underlying concept: generation time refers to the average time between transmission the virus from an

**Table 2. Sub-group analyses across study characteristics to explore the sources of heterogeneity of the estimated coronavirus's reproductive number.**

| Characteristics | Number of Reportings** | R (95% CI) | P | |
|---|---|---|---|---|
| | | | Heterogeneity | Meta-regression |
| **Country** | | | | |
| China | 14 | 3.14 (2.40–4.09) | <0.01 | <0.01 |
| China and overseas | 3 | 2.90 (2.78–3.02) | 0.490 | |
| Outside of China | 2 | 4.56 (2.27–9.17) | <0.01 | |
| Japan | 1 | 2.60 (2.41–2.81) | NA | |
| Diamond Princes Cruise ship, Japan | 2 | 2.71 (1.33–5.52) | 0.290 | |
| South Korea | 3 | 0.76 (0.34–1.70) | <0.01 | |
| Italy | 1 | 3.27 (3.16–3.38) | NA | |
| Germany | 1 | 6.07 (5.51–6.69) | NA | |
| Spain | 2 | 3.56 (1.62–7.82) | <0.01 | |
| France | 1 | 6.32 (5.72–6.99) | NA | |
| Global Initiative on Sharing Al Influenza Data | 1 | 2.10 (1.52–2.90) | NA | |
| **Method considered** | | | | |
| MLE | 4 | 2.63 (2.18–3.18) | <0.01 | <0.05 |
| EGR | 9 | 3.67 (2.91–4.64) | <0.01 | |
| SEIR | 6 | 1.97 (1.14–3.40) | <0.01 | |
| MCMC | 3 | 4.57 (2.68–7.78) | <0.01 | |
| NGMA | 3 | 4.36 (1.94–9.76) | 0.280 | |
| Others | 6 | 2.11 (1.60–2.79) | <0.01 | |
| **Model considered** | | | | |
| SEIR model | 11 | 2.81 (1.83–4.31) | <0.01 | 0.5216 |
| SIR model | 2 | 2.51 (2.05–3.08) | <0.01 | |
| Epidemic curve | 18 | 3.04 (2.60–3.55) | <0.01 | |
| **Number of cases** | | | | |
| ≤3162 | 16 | 2.51 (1.91–3.28) | <0.01 | 0.7758 |
| >3162 | 15 | 3.27 (2.47–4.31) | <0.01 | |
| **Type of reproductive number** | | | | |
| Basic reproductive number ($R_o$) | 32[a] | 3.17 (2.62–3.84) | <0.01 | 0.2047 |
| Net reproductive number ($R_e$) | 12[b] | 1.95 (1.63–2.34) | <0.01 | |
| Time-dependent reproductive number ($R_t$) | 6[c] | 4.42 (3.05–6.40) | <0.01 | |

Note: ** Number of studies 29 with reproductive number record 32 times (one study reported estimate for four different countries).

[a] Total 24 studies reported 32 different Ro,

[b] total 6 studies reported 12 different Re and

[c] total 3 studies reported 6 different Rt.

infected person to the non-infected person whereas serial interval refers duration between onset of symptoms in an index case to the transmission in a secondary case [65, 66, 68]. Moreover, the estimated reproductive number generated by mathematical models is dependent on numerous decisions made by the researcher such as homogeneity or heterogeneity of the population considered; use a deterministic or stochastic approach and which distributions to be used to describe the probable values of parameters [57].

We found the type of reproductive number considered was another important source of heterogeneity of the estimated reproductive number. For instance, this study found the summary of the basic reproductive number was 1.95, around half of the summary of the estimated time-dependent reproductive number (4.42). Three previous meta-analyses found the

**Table 3. Narrative synthesis of the studies included in the review.**

| Author, Study's Location | Model | Time/ period | Assumptions and method | Results |
|---|---|---|---|---|
| **T. Zhou et al, 2020 [6], China[b]** | SEIR model | before 26th Jan 2020 | With generation time of 8.4 days and 10 days and using the exponential growth rate method | Estimated basic reproductive number was varied from 2.83 to 3.34 (for 8.4 days generation time) or 3.28 to 3.93 (for 10 days generation time). |
| **Tang et al., 2020 [18], China[b]** | SEIR model (with isolation, quarantined) | 31 Dec 2019 to 15th Jan 2020 | The incubation period was 7 days, ignoring the asymptomatic infection in the model and using the next generation matrix approach | The estimated reproductive number was 6.47 (5.71–7.23) during the control measures of isolation and quarantine are implementing. |
| **T.-M. Chen et al., 2020 [22], Wuhan, China[b]** | SEIR (Bat-Host-Reservoir-People network model) | 10th Jan to 24th Jan 2020 | Assuming the mean incubation 5.2 days, mean infectious period 5.8 days and using the next generation matrix approach | The basic reproduction number estimated was 2.30 from the reservoir to person. It was increased to 3.58 when reached person-to-person level transmission. |
| **W. Zhou et al., 2020 [21], China[b]** | SEIHR model extended by quarantined | before 10 Jan 2020 | Parameterizing cumulative cases, deaths, the daily number of media reports and proportion of quarantined exposed by the virus and the estimation method was the next generation matrix approach | The basic reproductive number was 5.32. |
| **Rocklov et al., 2020 [30], Diamond Princess Cruise ship, Japan[b]** | SEIR model | 21 Jan to 19 Feb 2020 | The individual can mix randomly, the infectious period was 10 days and the contact rate were the same as early outbreak using the SEIR method. | The basic reproductive number was 14.80 without any intervention by using 79% infected persons in the ship. However, isolation and quarantine before 62.35% infected cases reduce this number to 1.78. |
| **D'Arienzo & Coniglio, 2020 [35], Italy[b]** | SIR model | 25 Feb to 12 Mar 2020 | Nearly everyone in Italy was considered as susceptible using the general SEIR method | The basic reproductive number was 3.10 while the number varies from 2.46 to 3.09 in different region across Italy. |
| **Najafi et al., 2020 [47], Western Iran[b]** | Infector-Infectee model | 22 Feb to 9 Apr 2020 | The Weibull distribution provides the best fit for GT and the mean 5.71 days and SD 3.89 days | The time-dependent reproductive number varied from 0.79 to1.88 for 7-day and from 0.92 to 1.64 for 14-day time-lapse. The decreasing trend inverses in April for both 7- and 14-day time-lapses. |
| **Wahaibi et al., 2020 [48], Oman** | Infector-Infectee model | 24 Feb to 03 Jun 2020 | Median SI is estimated 6 with inter-quartile range 3–14 that follow the gamma distribution. | The time-dependent reproductive number decreased from 3.70 (2.80–4.60) in mid-March to 1.30 (1.20–1.50) in late April 2020 due to non-pharmaceutical interventions. |
| **Al-Raeei, 2020 [49], Different countries** | SIR model | Upto 30 July 2020 | Based on the estimated coefficient of infection, recovery and mortality. | The basic reproductive number varies from 1.00 to 2.79 in different countries. |
| **Sarkar et al., 2020 [50], India** | SEIR model | Upto 30 April 2020 | Used next-generation matrix model | The average estimated basic reproductive number was 2.05. |
| **Aldila et al., 2020 [51], Indonesia** | SEIR model | 03 Mar to 10 Apr 2020 | Population is mixed homogeneously. | The basic reproductive number was reduced to 1.22 after implementation of movement control order (MCO) from 1.75. |
| **Bagal et al., 2020 [52], India** | SIR model | 22 Jan to 31 May 2020 | Lockdown protocol homogeneously implemented across the country | The net-reproductive number was estimated at 1.37. |
| **Ullah and Khan, 2020 [53], Pakistan** | SEIR model | 01 Mar to 31 May 2020 | Hospitalized people can transmit after interacting with the general susceptible people | The average estimated basic reproductive number was 1.87 |

Note: Studies included in the meta-analysis were summarized in Table 1.

summary estimate of the basic reproductive number ranged from 3.15 and 3.38 [12–14]. The sources of such heterogeneity are the underlying assumptions and the period between the initial infection and date of estimation [65, 69].

This study was first of its kind that provides an estimation of reproductive numbers based on the global literature. Moreover, we have considered the heterogeneity of the reproductive numbers estimated worldwide and explored the sources of heterogeneity across the characteristics of the selected papers. However, many other factors may explain the sources of

heterogeneity of the reported reproductive number of coronavirus worldwide which was not explored in this study because of the lack of data.

## Conclusion

The estimated summary reproductive number was 2.87. We found evidence of higher heterogeneity of the reproductive number reported worldwide. We found the country for which the reproductive number was estimated and the method that was used to estimate the reproductive number were significant for such heterogeneity. Our analyses indicate the possibility of a significant increase of coronavirus infections in near future. Strengthening existing preventive measures, as well as new policies and programs, are important to reduce new infections.

## Supporting information

**S1 File.**
(DOCX)

## Acknowledgments

The authors are grateful to the authors of the paper included in this review.

## Author Contributions

**Conceptualization:** Md. Arif Billah.

**Data curation:** Md. Arif Billah, Md. Mamun Miah.

**Formal analysis:** Md. Nuruzzaman Khan.

**Methodology:** Md. Nuruzzaman Khan.

**Software:** Md. Nuruzzaman Khan.

**Supervision:** Md. Nuruzzaman Khan.

**Writing – original draft:** Md. Arif Billah, Md. Nuruzzaman Khan.

**Writing – review & editing:** Md. Mamun Miah, Md. Nuruzzaman Khan.

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
