## [Decision Letter · Decision Letter 0]

13 Sep 2020

PONE-D-20-16375

Reproductive number of COVID-19: A systematic review and meta-analysis based on global level evidence

PLOS ONE

Dear Dr. Khan,

Thank you for submitting your manuscript to PLOS ONE. After careful consideration, we feel that it has merit but does not fully meet PLOS ONE’s publication criteria as it currently stands. Therefore, we invite you to submit a revised version of the manuscript that addresses the points raised during the review process.

In addition to the comments raised by the Reviewer, there are other issues that need to be addressed. Specifically:

1. Given that the authors are approaching a meta-analysis on such a specific topic, they should have made it clear that COVID-19 is not the virus (which is, indeed, called SARS-CoV-2). COVID-19 stands for Coronavirus disease-19, and is the disease caused by the virus, so the basic reproduction number cannot be referred to the disease, but to the etiological agent. Please do correct everywhere in the title, abstract, manuscript, tables and figures, where appropriate. No additional revision process can be carried on in the absence of this basic requirement.

2. The data search dates back to April 10, 2020. Given the rapidly evolving epidemiologcal scenario, the search needs to be updated to eventually include any new published study.

3.  "Data extraction and Analysis" paragraph, page 8, lines 4-5: "For the studies where more than one reproductive number reported based on different assumptions, we calculated the average reproductive number. This calculated average reproductive number was then used to give summary estimate". This approach is not entirely correct. When multiple estimates are available for the same study, a pooled estimate should be computed from the available data using a fixed-effect meta-analysis. Then the resulting estimate should be included in the overall meta-analysis (see as an example Flacco et al Heart 2020; heartjnl-2020-317336).

4. Table 2: some results from a meta-regression are listed, but they are not commented in the results, nor reported in the methodology: please do clarify.

5. Page 7, last line: "Meta-analysis [WAS] then used to give an average estimate of the reproductive number". The estimate resulting from a meta-analytical approach is not simply an average, and should not be defined as such. It is a pooled, or summary estimate. Please do correct everywhere in the text where appropriate.

6.  At least two previous meta-analyses were published on the same topic (He W et al. J Med Virol. 2020 May 29:10.1002/jmv.26041. doi: 10.1002/jmv.26041*; *Alimohamadi Y et al. J Prev Med Public Health. 2020 May;53(3):151-157. doi: 10.3961/jpmph.20.076). Please discuss your results also in light of these previous findings.

7. Minor issues:

- The length of the introduction could be easily reduced, as many of the reported information on the hystory of the pandemic and on the main characteristic of the SARS-CoV-2 virus are already well known, and could be reduced to one or two short paragraphs. Please go straight to the point.

- The manuscript needs a careful language editing, as there are some errors and misspellings (see as an example: "Meta-analysis WAS then used to give an average estimate of the reproductive number.", or "the estimated reproductive number was around double (..) in outside of China than China").

Only if the authors are able to address the above mentioned issues, in addition to those raised by the Reviewer, the manuscript may be reconsidered for publication.

We look forward to receiving your revised manuscript.

Kind regards,

Maria Elena Flacco, M.D.

Academic Editor

PLOS ONE

Journal Requirements:

5. We note that this manuscript is a systematic review or meta-analysis; our author guidelines therefore require that you use PRISMA guidance to help improve reporting quality of this type of study. Please upload copies of the completed PRISMA checklist as Supporting Information with a file name “PRISMA checklist”.

Reviewers' comments:

Reviewer's Responses to Questions

**Comments to the Author**

1. Is the manuscript technically sound, and do the data support the conclusions?

Reviewer #1: Yes

2. Has the statistical analysis been performed appropriately and rigorously? 

Reviewer #1: Yes

3. Have the authors made all data underlying the findings in their manuscript fully available?

Reviewer #1: Yes

4. Is the manuscript presented in an intelligible fashion and written in standard English?

Reviewer #1: Yes

5. Review Comments to the Author

Reviewer #1: Abstract:

Aim and study methodology is fairly stated but need to touch on some special feature on the topic to provoke read full paper.

Introduction:

Introduction could have been more effective. Context of the study is moderately stated but lacks enough logical progression. Importance of this study should benefit the readers and provide a rough solution which will be convinced in discussion section.

Please refer to the previous meta-analyzes(for example: Estimate of the Basic Reproduction Number for COVID-19: A Systematic Review and Meta-analysis. Alimohamadi, Yousef et al. J Prev Med Public Health 2020; 53(3): 151-157. DOI: https://doi.org/10.3961/jpmph.20.076) and mention the relative advantages of your work over previous studies.

Figure 1 needs to be self explanatory with stand alone graphics. a and b are not shown in the fig.

Methodology:

Selection, presentation and sampling of various studies were fairly explained. As you mentioned in the introduction section there are at least 3 types of R, but according to the search strategy, it is not clear which type of R has been studied. Has subgroup analysis been performed for different types of R? If yes, why aren't the results presented?

If possible, some insights from the authors about possible reasons for those larger outliers (around 6), or any particular characteristics of those studies (e.g. small sample, earlier stage, or particular region?)

Results:

Figures and tables needs to be self explanatory with stand alone graphics.

Please provide the results of the quality assessment of each study in the table 1.

Please provide the detailed results of the funnel plot , Egger’s test and Trim-and Fill procedure.

6. PLOS authors have the option to publish the peer review history of their article (what does this mean?). If published, this will include your full peer review and any attached files.

Reviewer #1: No

---

## [Author Response · Author response to Decision Letter 0]

7 Oct 2020

Added in a MS word file named "Response to Reviewers"

---

## [Decision Letter · Decision Letter 1]

28 Oct 2020

Reproductive number of coronavirus: A systematic review and meta-analysis based on global level evidence

PONE-D-20-16375R1

Dear Dr. Khan,

We’re pleased to inform you that your manuscript has been judged scientifically suitable for publication and will be formally accepted for publication once it meets all outstanding technical requirements.

Kind regards,

Maria Elena Flacco, M.D.

Academic Editor

PLOS ONE

Additional Editor Comments (optional):

Reviewers' comments:

Reviewer's Responses to Questions

**Comments to the Author**

1. If the authors have adequately addressed your comments raised in a previous round of review and you feel that this manuscript is now acceptable for publication, you may indicate that here to bypass the “Comments to the Author” section, enter your conflict of interest statement in the “Confidential to Editor” section, and submit your "Accept" recommendation.

Reviewer #1: All comments have been addressed

2. Is the manuscript technically sound, and do the data support the conclusions?

Reviewer #1: Yes

3. Has the statistical analysis been performed appropriately and rigorously? 

Reviewer #1: I Don't Know

4. Have the authors made all data underlying the findings in their manuscript fully available?

Reviewer #1: No

5. Is the manuscript presented in an intelligible fashion and written in standard English?

Reviewer #1: Yes

6. Review Comments to the Author

Reviewer #1: (No Response)

7. PLOS authors have the option to publish the peer review history of their article (what does this mean?). If published, this will include your full peer review and any attached files.

Reviewer #1: No

---

## [Editor Report · Acceptance letter]

30 Oct 2020

PONE-D-20-16375R1 

Reproductive number of coronavirus: A systematic review and meta-analysis based on global level evidence 

Dear Dr. Khan:

I'm pleased to inform you that your manuscript has been deemed suitable for publication in PLOS ONE. Congratulations! Your manuscript is now with our production department. 

Kind regards, 

on behalf of

Dr. Maria Elena Flacco 

Academic Editor

PLOS ONE